# Effects of L-valine in layer diets containing 0.72% isoleucine

Usman Liaqat[1], Yasir Ditta[1]*, Saima Naveed[1], Annie King[2]*, Talat Pasha[1], Sana Ullah[1], Khalid Abdul Majeed[1]

**1** Department of Animal Nutrition, University of Veterinary and Animal Sciences, Ravi Campus, Pattoki, Pakistan, **2** Department of Animal Science, University of California Davis, Davis, CA, United States of America

* yasirad787@gmail.com (YD); ajking@ucdavis.edu (AJK)

**Data Availability Statement:** All relevant data are within the Dryad repository at: https://doi.org/10.5061/dryad.v41ns1rx2.

**Funding:** This work was supported by AMINO LAB®, ISO 9001, Evonik, Singapore. The funders

## Abstract

In a previous study with LSL-LITE layers (-23 to 30-week-old), isoleucine at 0.72% and 0.84% produced values for FCR at 1.45 and 1.44, respectively and shared significance with 0.78% isoleucine (1.49). Considering that FCR is an important standard in the poultry industry due to the cost for adding feed ingredients such as synthetic amino acids and the low FCR of 1.45, 0.72% isoleucine was chosen for further study with LSL-LITE layers (n = 490 at 33- to 40-week-old) to determine effects on production and egg quality. The study included 7 diets (2730 Kcal kg metabolizable energy and constant isoleucine at 0.72%) containing varying quantities of valine [0.72 (Control), 0.75, 0.78, 0.81, 0.84, 0.87 or 0.90%] x 7 replicates x 10 hens/replicate. Significance at $P \leq 0.05$ and $P < 0.10$ was determined. Level and week were significant for feed intake, egg production, and FCR; the interaction of level x week (L*W) was significant for feed intake and FCR. An isoleucine:valine of 1.233 corresponding to 0.72% isoleucine and 0.87% valine produced the lowest FCR of 1.30 (a 2.26% decrease compared to the Control at 1.33 ± 0.04). All measurements for external egg quality, except shape index and eggshell thickness, were significant for level. Week was significant for all parameters except shell thickness; L*W was significant for external quality measurements except shape index and shell thickness. Level, week, and L*W were significant for internal egg quality measurements. Serum protein and H1 titer were significant for level. Various production, egg quality, and biochemical measurements were significantly different from the control (0.72% isoleucine and 0.72% valine) at 0.81 to 0.87% valine. Findings of this study will aid researchers and commercial producers in narrowing the range of isoleucine, valine, and leucine needed for effects on particular parameters. Knowledge gained from this and others studies will eventually lead to an understanding of synergistic and antagonistic effects of branched chain amino acids in feed for various genetic types of layers throughout their productive lifetime.

had no role in study design, data collection and analysis, decision to publish, or preparation of the manuscript.

**Competing interests:** The authors of the manuscript have no competing interests.

## Introduction

Branched chain amino acids (BCAAs) are precursors of other amino acids and proteins. Metabolized extra-hepatically, they are energy precursors and aid in repair of muscle proteins (Brosnan and Margaret, 2006) [1]. It has been assumed that leucine requirements can generally be met by various sources of protein in diets. Thus, research has focused on appropriate amounts of isoleucine and valine in the diet to maintain antioxidant capacity, gut immunity, and critical metabolic processes such as fatty acid metabolism (Azzam et al., 2015; Dong et al., 2016; Wen et al., 2019; Bai et al., 2021) [2–5].

In layers, isoleucine is said to be the fourth limiting amino acid after tryptophan (Harms and Ivey, 1993) [6]. Several studies have focused on isoleucine requirements for young hens. Harms and Russell (2001) provided 0.49 to 0.61% isoleucine to Hy-Line W36 (36- to 44-week-old) diets containing supplemental amino acids to ensure that isoleucine was first limiting [7]. Egg production, weight, and contents were significantly increased by all isoleucine additions. Results revealed daily isoleucine requirement of 589.2, 601.2, and 601.4 mg/day for egg production, weight, and content, respectively, indicating 12.6 mg isoleucine/g of egg content. A follow-up experiment for Hy-Line layer (35 to 43 week-old) diets with 0.39 to 0.61% isoleucine indicated significantly elevated egg mass, egg production, and egg weight for levels above 0.51% (Shivazad et al., 2002) [8]. Egg mass, egg weight, and egg production decreased with decreasing isoleucine. Feed consumption, energy intake, and body weight were significantly decreased at 0.45% isoleucine. Broken-line regressions showed requirements of 449.8, 497.0, and 469.0 mg isoleucine/day for egg production, weight, and mass, respectively, corresponding to 9.30 mg isoleucine/g egg mass. Parenteau et al. (2020) determined responses to isoleucine supplementation in low crude protein diets of laying hens (20- to 27 and 28- to 46- week-old) [9]. Findings showed that a reduction of crude protein by 2% was possible when diets were fortified with synthetic amino acids (methionine, lysine, threonine, and tryptophan) + isoleucine. Optimum responses occurred at 82 and 88% standardized ileal digestible isoleucine:lysine [9].

Results of earlier studies reported dietary valine requirements between 0.54 and 0.72% for maximum layer performance (Johnson and Fisher, 1958; Hurwitz and Bornstein, 1978) [10, 11]. Other results suggested from 610 to 786 mg valine/hen/day (McDonald and Morris, 1985; Jensen and Colnago, 1991) [12, 13]. NRC (1977) [14] recommended 550 mg/hen/day [15]. Later the recommendation was 600 mg/hen/day, followed by a final recommendation (to date) of 700 mg/hen/day (NRC,1994) [15]. As with isoleucine, some research has focused on valine requirements for layers of young to medium age (Harms and Russell, 2001) [16]. Hy-Line W-36 layers (39- to 47-week-old) were fed diets containing 0.525 to 0.700% valine. Egg production and egg weight were increased by 0.630% and 0.655% valine, respectively. Daily valine requirement of 592.5, 677.7, and 619.0 mg/hen/day were reported for improved egg production, weight, and contents respectively. A value of 13.1 mg valine/g of egg content was indicated by broken-line regression [16]. In work by Bregendahl et al. (2008) with Hy-Line W-36 hens (26- to 34-week-old), the digestible amino acid requirement used to calculate the ideal amino acid ratio for maximum egg mass was 501 mg/day of valine. The ideal amino acid ratio for maximum egg mass was 93% valine relative to lysine (100%) [17].

A critical review of amino acids requirements for laying hens was recently provided by Macelline et al. (2021) [18]. Values from producers of five layer breeders ranged from 640 to 760 and 700 to 819 mg/bird/day for isoleucine and valine, respectively (Macelline et al., 2021) [18]. Values for ideal digestible amino acid ratios for layers ranged from 79 to 86 and 88 to 102 mg/bird/day for isoleucine and valine, respectively (Macelline et al., 2021) [18].

As antagonists, isoleucine and valine, branched chain amino acids (BCAA's) share the same pathway to cross the cellular membrane, are degraded by the same enzymes, and cross the

blood-brain barrier at the same time (Benton et al., 1956; Rogers et al., 1993; Harper, 1984) [19–21]. As more synthetic amino acids are used in poultry diets, research will continue to ensure that complementary quantities of BCAA's are maintained. The ideal amino acid ratio as determined by the Goettingen approach was lysine, 100; isoleucine, 75; and valine, 90. As determined by the Louvain approach, it was lysine, 100; isoleucine, 80; and valine, 91 (Soares et al., 2018) [22].

In a previous study with LSL-LITE layers (23- to 30-week-old), 0.72% and 0.84% total iso-leucine in diets produced low FCR of 1.45 and 1.44, respectively, and shared significance of FCR (1.49) with isoleucine at 0.78% (Ullah et al., 2021) [23]. The corresponding ratios of lysine: isoleucine for these percentages were 0.72% = 1:0.867; 0.78% = 1:0.939, and 0.84% = 1:0.988. Due to the low FCR and cost consideration when adding synthetic isoleucine, 0.72% was designated for further study with various concentrations of valine to determine effects for layer performance as well as external and internal egg quality in LSL-LITE layers (33- to 40-week-old).

## Results and discussion

The standard error of the mean (SE) is pooled for each individual measurement for produc-tion; external egg quality; internal egg quality; and serum biochemisty, digestibility, and titers. SE is included with the value for the Control of each parameter discussed below.

### Production (Table 1)

Increasing valine had linear ($p < 0.01$), quadratic ($p = 0.01$), and cubic ($p < 0.01$) effects on feed intake. Except for 0.87%, all levels of valine significantly increased feed intake. At 0.78% valine, feed intake (111.52 kg) was 6.33% greater than the Control (104.88 kg ± 3.18). Lelis et al. (2014) also reported a linear increase in feed intake when feeding brown layers (42- to 54-wk-old) valine to lysine ratios of 84 to 100%, compared to our ratio of ~94 to 100% [24]. They suggested an ideal digestible valine to lysine ratio of 92% (0.607% digestible valine or 567 mg/hen/day of digestible valine) [24]. Differences in age and genetics of layers may explain dif-ferences in findings. Week (fluctuating values) and L*W were significant (p < 0.01) for feed intake in the present study.

Level produced a quadratic *(p = 0.07)* trend and a cubic ($p < 0.01$) effect for production of isoleucine:valine at 1.048 and 1.192, respectively.

The highest production (96.85% for 0.87% valine) produced an increase in weight of 2.77% compared to the Control (94.24% ± 2.01). Lelis et al. (2014) also noted that increasing ratios of valine:lysine had a quadratic effect on egg production of brown layers [24]. When Harms and Russell (2001) fed HyLine W-36 layers (39- to 47-week-old) diets containing 0.525 to 0.700% valine, egg production was increased at 0.630% [16]. Azzam (2015) found no effect on produc-tion when feeding 0.1 to 0.4% to Hyline Brown hens (40- to 47- week old) [2]. Results of Azzam (2015) are likely different from others due to the lower quantities (0.1 to 0.4%) of valine provided [2]. Age and breed of layers are likely important factors for differences in results as well.

For FCR, level was significant ($p < 0.01$); there were linear ($p < 0.03$) and cubic ($p < 0.01$) effects. A low feed intake and the highest egg production at 0.87% valine produced the lowest FCR (1.30) compared to the Control (1.33 ± 0.04). Lelis et al. (2014) reported a quadratic effect for FCR [24]. Azzam (2015) reported no effect on FCR with an increase of valine, likely due to the lower quantities fed [2]. Week was significant *(p = 0.01)* as values fluctuated over time. A trend *(p = 0.10)* was observed for L*W.

**Table 1. Valine level and production parameters[@] of 33- to 40-week-old hens.**

| Level (%) | Lys:val[#] | Iso:val[##] | Feed intake (kg) | Egg production (%) | FCR[$] |
|---|---|---|---|---|---|
| Control[$ $] | 0.867 | 0.986 | 104.88[d] | 94.24[c] | 1.33[b] |
| 0.75 | 0.900 | 1.027 | 108.37[bc] | 95.41[cb] | 1.33[b] |
| 0.78 | 0.939 | 1.068 | 111.52[a] | 95.41[cb] | 1.37[a] |
| 0.81 | 0.975 | 1.109 | 109.59[b] | 95.55[b] | 1.35[ab] |
| 0.84 | 1.010 | 1.151 | 109.64[b] | 94.31[c] | 1.36[a] |
| 0.87 | 1.048 | 1.192 | 104.32[d] | 96.85[a] | 1.30[c] |
| 1.048 | 11.084 | 11.233 | | | |
| 0.90 | 1.084 | 1.233 | 107.27[c] | 95.39[cb] | 1.35[ab] |
| Pooled SE | | | 3.18 | 2.01 | 0.04 |
| **Week (age)** | | | | | |
| 33 | | | 116.74[a] | 93.85[b] | 1.38[a] |
| 34 | | | 107.97[c] | 94.89[ab] | 1.36[ab] |
| 35 | | | 106.22[cd] | 95.58[a] | 1.34[b] |
| 36 | | | 104.29[e] | 95.91[a] | 1.31[c] |
| 37 | | | 100.45[f] | 95.60[a] | 1.28[c] |
| 38 | | | 111.17[b] | 95.39[a] | 1.31[c] |
| 39 | | | 110.84[b] | 95.82[a] | 1.38[a] |
| 40 | | | 105.82[de] | 95.43[a] | 1.36[ab] |
| Pooled SE | | | 2.41 | 1.52 | 0.03 |
| Level | | | < 0.01 | < 0.01 | < 0.01 |
| Week | | | < 0.01 | 0.01 | < 0.01 |
| L*W[+] | | | <0.01 | [*] | 0.10 |
| Linear 00Linear | | | < 0.01 | 0.26 | 0.03 |
| Quadratic | | | 0.01 | 0.07 | 0.35 |
| Cubic | | | < 0.01 | < 0.01 | < 0.01 |

[@] Means (treatments × 7 replicates × 10 layers per replicate) with the same superscript within a column do not differ significantly at P ≤ 0.05.

[#] Lysine:valine.

[##] Isoleucine:valine.

[$] Feed consumed per kg eggs.

[$ $] Control containing isoleucine and valine, both at 0.72% of the diet.

[+] Interaction of level by week.

[*] Data unavailable.

## External egg quality (Table 2)

External egg quality parameters are shown in Table 2. Level was significant for all external egg quality measurements except shape index and shell thickness where there were trends at $p = 0.08$ and $p = 0.10$, respectively. Egg weight trended linearly ($p = 0.09$) with quadratic ($p = 0.04$) and cubic ($p = 0.008$) responses. Compared to the control (60.73 g ± 0.42), a significant peak (61.61g) for weight occurred at 0.84% valine.

Harms and Russell (2001) noted that for Hy-Line W-36 layers (39- to 47-week-old), egg weight started to increase at 0.655% valine when layers received 0.525 to 0.700% valine.16 Lelis et al. (2014) fed Dekalb Brown layers (42- to 54-week-old) while Azzam (2015) fed Hyline Brown layers (40- to 47-week-old) and found no effect of valine for egg weight or specific gravity.2, 24 Information for our study with SL LITE layers, those of Lelis et al. (2014) and Azzam (2015) with brown layers suggest a thorough investigation of the literature for effects of valine on egg quality traits form various genetic types of layers in the same age range 2, 24.

**Table 2. Valine level and external egg quality[@] of 33- to 40-week-old hens.**

| Level (%) | Lys: val[#] | Iso: val[##] | Egg weight (g) | Specific gravity | Shape index | Shell weight (g) | Shell thickness (mm) | Prop shell weight[$] |
|---|---|---|---|---|---|---|---|---|
| Control[*] | 0.867 | 0.986 | 60.73[b] | 1.0739[d] | 74.17[b] | 6.61[a] | 0.33[b] | 10.81[ab] |
| 0.75 | 0.900 | 1.027 | 60.72[b] | 1.0772[bc] | 74.29[ab] | 6.45[ab] | 0.36[a] | 10.60[b] |
| 0.78 | 0.939 | 1.068 | 60.44[c] | 1.0779[b] | 74.60[ab] | 6.59[a] | 0.34[b] | 10.84[ab] |
| 0.81 | 0.975 | 1.109 | 60.17[d] | 1.0802[a] | 74.77[a] | 6.56[ab] | 0.34[b] | 10.95[a] |
| 0.84 | 1.010 | 1.151 | 61.61[a] | 1.0760[c] | 74.06[b] | 6.39[b] | 0.33[b] | 10.31[c] |
| 0.87 | 1.048 | 1.192 | 60.58[bc] | 1.0779[b] | 74.20[ab] | 6.46[ab] | 0.34[b] | 10.61[b] |
| 0.90 | 1.084 | 1.233 | 60.55[bc] | 1.0765[bc] | 74.47[ab] | 6.56[ab] | 0.34[b] | 10.86[ab] |
| Pooled SE | | | 0.42 | 0.002 | 0.98 | 0.29 | 0.039 | 0.46 |
| **Week (age)** | | | | | | | | |
| 33 | | | 59.05[e] | 1.08[b] | 74.71[ab] | 6.00[e] | 0.34[abc] | 10.20[d] |
| 34 | | | 60.06[d] | 1.09[a] | 74.39[abc] | 6.25[d] | 0.34[abc] | 10.32[cd] |
| 35 | | | 60.62[c] | 1.07[c] | 73.92[c] | 6.25[d] | 0.35[ab] | 10.27[d] |
| 36 | | | 61.00[b] | 1.07[d] | 73.95[c] | 6.48[c] | 0.33[c] | 10.59[c] |
| 37 | | | 61.08[b] | 1.10[e] | 74.87[a] | 6.91[ab] | 0.33[bc] | 11.29[a] |
| 38 | | | 61.18[ab] | 1.07[e] | 74.14[bc] | 7.01[a] | 0.33[bc] | 11.53[a] |
| 39 | | | 61.40[a] | 1.07[f] | 74.43[bac] | 6.47[c] | 0.34[abc] | 10.58[c] |
| 40 | | | 61.09[b] | 1.08[b] | 74.53[abc] | 6.76[b] | 0.36[a] | 10.90[b] |
| Pooled SE | | | 0.31 | 0.001 | 0.74 | 0.21 | 0.03 | 0.35 |
| Level | | | < 0.01 | < 0.01 | 0.08 | 0.04 | 0.10 | < 0.01 |
| Week | | | < 0.01 | < 0.01 | < 0.01 | < 0.01 | 0.08 | < 0.01 |
| L*W[**] | | | 0.01 | < 0.01 | 0.06 | < 0.01 | 0.21 | 0.03 |
| Linear | | | 0.09 | < 0.01 | 0.22 | 0.29 | 0.02 | 0.43 |
| Quadratic | | | 0.04 | < 0.01 | 0.60 | 0.18 | 0.15 | 0.15 |
| Cubic | | | 0.008 | 0.70 | 0.20 | 0.03 | 0.16 | 0.01 |

[@] Means (7 treatments × 7 replicates × 10 layers per replicate) with the same superscript within a column do not differ significantly at P ≤ 0.05.

[#] Lysine:valine.

[##]Isoleucine:valine.

[$]Proportional shell weight, %

[*]Control as isoleucine and valine, both at 0.72% of the diet.

[**]Interaction of level by week.

Level was linear and quadratic (both at *p < 0.001*) for specific gravity while egg shell weight was cubic (*p = 0.03*). Specific gravity peaked (1.0802) at 0.81% valine compared to the Control (1.0739 ± 0.002). Specific gravity and shell weight are closely related because specific gravity is the relationship between the shell weight and other egg components (Butcher and Miles, 2017) [25]. However, shell weight was not linear but cubic (*p = 0.03*). Shell weight was different from the Control (6.61 g + 0.29) at 0.84% valine (6.39 g), producing 3.33% less shell.

Increasing valine produced a trend for shell thickness at *p = 0.10* and a linear effect (*p = 0.02*). Shell thickness increased at 0.75% valine (0.36 mm), producing 9.09% greater thickness compared to the Control (0.33 mm ± 0.039 SE). L*W was not significant for shell thickness (*p = 0.21*) and week produced a linear trend at *p = 0.08* for shell thickness. The trend for shell thickness by level may have been confounded by a trend for week associated with deposition of minerals. This supposition was supported by findings of Park and Sohn (2018) who conducted a study to examine the histological change of the eggshell endometrium for 30-, 60-, and 72-wk-old commercial layers [26]. As well, they analyzed eggshell ultrastructure and ionic composition. These investigators noted that eggshell weight significantly increased

between 30 (6.72 g ± 0.026) and 60 weeks (7.08 g ± 0.40). $Na^+$, $K^+$, and $V^{2+}$ increased with age while $CA^{2+}$, $Co^{2+}$, and $S^{2-}$ significantly decreased. Ultimately, egg shell structure was affected causing increased weakness of the shell while thickness and density did not change.

Level of valine was significant ($p < 0.01$) for proportional shell weight and there was a cubic effect at $p = 0.01$. The lowest proportional shell weight occurred at 0.84% valine (10.31%), compared to the Control (10.82% + 0.42). The reduction was 4.63%. When eggs have similar weights, the amount of shell can cause a difference in specific gravity (Butcher and Miles, 2017) [25]. At 0.84% valine, egg weight was highest (61.61 g), proportional shell weight was lowest (10.81%); however specific gravity, while at the numerically lowest point (10.61), shared significance with other values. For proportional shell weight, week was significant at $p < 0.01$ and $L^*W$ was significant at $p = 0.03$.

Shape index for level ($p = 0.08$) and $L^*W$ ($p = 0.06$) were not highly significant. The importance of shape index as associated with shell durability is controversial. While some investigators noted that an index >76 is needed for shell durability during marketing, others show no correlation for these measurements (Sarica and Erensayin, 2004; Duman et al., 2016) [27, 28]. Unless eggs shells are extremely fragile due to feed source, disease, or handling, shape index may not affect sale of eggs in local farmers' markets or for consumers' use from backyard production.

**Internal egg measurements (Table 3).** Level was significant for all internal measurements (Table 3). Albumen and yolk indices were linear, quadratic and, cubic at $p = <0.01$. The Control for the albumen index was 7.57 ± 0.38. It increased by 8.59% and 10.30% for 0.84% valine (8.22) and 0.90% valine (8.35), respectively. The yolk index for the Control (42.61 + 0.96) increased by 5.09% with valine (44.78) at both 0.75% and 0.78%. Haugh unit for the Control (93.32 HU ± 1.87) had the greatst increased of 2.90% at 0.84% valine (96.02 HU) and 3.24% at 0.90% valine (96.33 HU).

The yolk:albumen had quadratic ($p = 0.013$) and cubic effects ($p = 0.01$) for level. The Control value of 0.43 ± 0.03 for yolk:albumen decreased for hens fed 0.84% (0.40) and the reduction was 6.98%. The albumin index and Haugh unit exhibited a general downward trend as hens aged, $r^2$ for these two variables was 0.95. These temporal effects likely confounded findings for level.

Proportional yolk weight had quadratic ($p < 0.01$) and cubic ($p = 0.03$) effects. The Control value was 26.90% ± 0.80 and there was a reduction of 4.46% for valine levels at 0.84% (25.70%). Level produced a linear trend for yolk color at $p = 0.07$ and a quadratic effect ($p = 0.02$).

For Lelis et al. (2014), egg weight and egg internal quality were not influenced by the different dietary digestible valine-to-lysine ratios [24]. Internal quality measurements associated with yolk at various isoleucine:valine used in the present study may be associated with synergistic or atagonistic effects on deposition of lipids influenced by branched chain amino acids and need further study (Bai et al., 2017) [5]. Yolk color may be affected by deposition of fat soluble carotenoids assoicated with effects of BCAA's.

As noted by Macelline et al. (2021), the higher values for amino acid ratios for layers used in our study and that for Lelis et al. (2014) were warranted [18, 24]. These levels may be especially important for layers to produce egg content (protein and lipids) during the early to middle range of egg production, maintain antioxidant capacity, gut immunity, and critical metabolic processes (Azzam et al., 2015; Dong et al., 2016; Wen et al., 2019; Bai et al., 2021) [2–5].

## Serum biochemistry and crude protein digestibility (Table 4)

No influence of level was noted for serum glucose. For serum albumin, there was a trend ($p = 0.06$) for level. Valine level was also significant ($p = 0.02$) for total protein due to

**Table 3. Valine level and internal egg quality[@] of hens at 33–40 weeks.**

| Levels (%) | Lys: val[#] | Iso: val[##] | Albumen index | Yolk index | Yolk color | Yolk: albumen | Haugh unit | Prop. yolk weight[$] |
|---|---|---|---|---|---|---|---|---|
| Control* | 0.867 | 0.986 | 7.57[c] | 42.61[d] | 5.22[ab] | 0.43[a] | 93.31[c] | 26.90[a] |
| 0.75 | 0.900 | 1.027 | 7.55[c] | 44.78[a] | 5.13[b] | 0.42[ab] | 93.94[c] | 26.54[abc] |
| 0.78 | 0.939 | 1.068 | 7.43[c] | 44.78[a] | 5.16[b] | 0.42[ab] | 93.34[c] | 26.40[bc] |
| 0.81 | 0.975 | 1.109 | 7.49[c] | 42.94[c] | 5.34 [a] | 0.43[a] | 93.75[c] | 26.80[ab] |
| 0.84 | 1.010 | 1.151 | 8.22[a] | 42.98[c] | 5.27[ab] | 0.40[c] | 96.02[a] | 25.70[d] |
| 0.87 | 1.048 | 1.192 | 8.01[b] | 42.78[dc] | 5.25[ab] | 0.41[bc] | 94.97[b] | 26.21[c] |
| 0.90 | 1.084 | 1.233 | 8.35[a] | 42.61[dc] | 5.35 [a] | 0.41[bc] | 96.33[a] | 26.60[abc] |
| Pooled SE | | | 0.38 | 0.96 | 0.300 | 0.03 | 1.87 | 0.80 |
| **Weeks (age)** | | | | | | | | |
| 33 | | | 8.32[a] | 43.00[b] | 3.32[g] | 0.41[c] | 96.58[a] | 26.04[c] |
| 34 | | | 8.02[bc] | 42.70[bc] | 4.1 [f] | 0.41[bc] | 95.86[a] | 26.19[c] |
| 35 | | | 8.05[b] | 43.54[a] | 5.27[e] | 0.40[c] | 95.67[a] | 25.99[c] |
| 36 | | | 7.54[d] | 42.95[b] | 5.90[b] | 0.42[abc] | 93.65[bc] | 26.42[bc] |
| 37 | | | 7.87[bc] | 44.09[a] | 6.45[a] | 0.43[ab] | 94.43[b] | 26.67[ab] |
| 38 | | | 7.79[c] | 43.90[a] | 5.44[d] | 0.42[abc] | 94.15[b] | 26.71[ab] |
| 39 | | | 7.44[d] | 42.8 1[b] | 5.80[bc] | 0.42[abc] | 93.00[c] | 26.69[ab] |
| 40 | | | 7.39[d] | 42.20[c] | 5.65 [c] | 0.43[a] | 92.84[c] | 26.95[a] |
| Pooled SE | | | 0.29 | 0.72 | 0.22 | 0.02 | 1.41 | 0.61 |
| Level | | | <0.01 | < 0.01 | 0.04 | < 0.01 | < 0.01 | < 0.01 |
| Weeks | | | <0.01 | < 0.01 | < 0.01 | < 0.01 | < 0.01 | < 0.01 |
| L*W** | | | < 0.01 | < 0.01 | < 0.01 | 0.04 | < 0.01 | 0.01 |
| Linear | | | < 0.01 | < 0.01 | 0.07 | 0.94 | 0.004 | 0.23 |
| Quadratic | | | < 0.01 | < 0.01 | 0.02 | 0.013 | < 0.01 | < 0.01 |
| Cubic | | | < 0.01 | 0.003 | 0.93 | 0.01 | < 0.01 | 0.03 |

[@] Means (7 treatments × 7 replicates × 10 layers per replicate) with the same superscript within a column do not differ significantly at P ≤ 0.05.

[#] Lysine:valine.

[##] Isoleucine:valine.

[$] Feed consumed/kg egg.

* Control as isoleucine and valine, both at 0.72% of the diet.

** Interaction of level by week.

fluctuation; there was no linear response. Valine at 0.81%, while not different from other values, produced a significantly lower value (3.22 g/dl) than that of the Control (4.30 g/dl ± 0.03) for total protein. The H9 titer did not respond to level of valine. The H1 titer for NDV was significant (*p = 0.02*) with a linear (*p < 0.01*) effect and a trend for a cubic response (*p = 0.07*). For level, there was a cubic (*p = 0.07*) trend and a linear effect (*p = 0.02*) for ileal protein digestibility.

Contrary to our results, Azzam et al. (2015) reported a significant quadratic effect on serum glucose for increasing valine that peaked at 0.2% but; there was no effect of valine level on total serum protein (Azzam, 2014) [2, 29]. The general downward response for total protein seemed to support findings of Bunchasak et al. (2005) [30] who found a significant response for decreased serum protein with increasing dietary valine (0.66, 0.76, and 0.85) in broilers [29]. However, Thornton et al. (2006) found insignificant effects of digestible valine on innate or adaptive immunity in broilers [31].

**Table 4. Effect of valine level[@] on serum biochemistry and ileal digestibility of crude protein in laying hens between 33–40 weeks.**

| Level (%) | Lys: val[#] | Ile: val[##] | Glucose (mg/dl) | Serum albumin (g/dl) | Total protein (g/dl) | H9 titer[$] | H1 titer for NDV[$$] | Digestability[+] |
|---|---|---|---|---|---|---|---|---|
| 0.72[*] | 0.867 | 0.986 | 148.29 | 2.90 | 4.30[a] | 8.57 | 8.42[ab] | 60.92[ab] |
| 0.75 | 0.900 | 1.027 | 151.43 | 3.47 | 3.75[ab] | 8.28 | 8.71[ab] | 72.88[a] |
| 0.78 | 0.939 | 1.068 | 146.86 | 2.88 | 3.42[ab] | 8.14 | 9.14[a] | 72.89[a] |
| 0.81 | 0.975 | 1.109 | 149.29 | 3.21 | 3.22[b] | 7.28 | 8.57[ab] | 74.04[a] |
| 0.84 | 1.010 | 1.151 | 142.71 | 3.28 | 3.42[ab] | 8.42 | 8.00[ab] | 67.88[ab] |
| 0.87 | 1.048 | 1.192 | 117.71 | 2.77 | 3.57[ab] | 7.57 | 7.71[b] | 52.84[b] |
| 0.90 | 1.084 | 1.233 | 146.14 | 2.64 | 3.37[ab] | 8.57 | 8.00[ab] | 66.11[ab] |
| Pooled SE | | | 13.33 | 0.28 | 0.30 | 0.53 | 0.41 | 6.90 |
| P value | | | 0.20 | 0.06 | 0.02 | 0.13 | 0.02 | 0.07 |
| Linear | | | 0.11 | 0.15 | 0.55 | 0.67 | < 0.01 | 0.02 |
| Quadratic | | | 0.82 | 0.77 | 0.43 | 0.41 | 0.957 | 0.57 |
| Cubic | | | 0.19 | 0.11 | 0.17 | 0.76 | 0.07 | 0.12 |

[@] Means (7 treatments × 7 replicates × 2 layers per replicate) with the same superscript within a column do not differ significantly at P ≤ 0.05.

[#] Lysine:valine.

[##] Isoleucine:valine.

[$] Detection for subtype of avian influenza.

[$$] Detection for Newcastle Disease.

[+] Ileal protein digestibility

[*] Control as isoleucine and valine, both at 0.72% of the diet.

Our significance for level and a linear response for NDV titer agreed with early results of Bhardava et al. (1969) who found a positivie effect for increasing high levels of valine (0.9 to 1.5%) on NDV titers in diets of 4-day-old chicks [32]. These findings are likely related to the requirement of valine for antibody production (killer-cell activity and lymphocyte proliferation) as noted by Calder (2006) for mammals [33]. Results from Jian et al. (2021) revealed a significant quadratic decrease in serum IgA and IgM with increasing valine (0.59, 0.64, 0.69, 0.74, and 0.79%) in diets of 33- to 40- week-old Fengda No.1 laying hens. Serum IgM levels did not change among diets [34]. Investigators (Azzam et al., 2015) reported that high levels of valine did not negatively affect the immune function in layers [2]. As noted by Kidd et al. (2021), the quantity of leucine in the diet must be considered when examining the effects of valine on responses in broilers [35]. Perhaps future work on immune responses in layers should always include responses across levels of the BCAA's simultaneously.

The supplemental amino acids, threonine, also boosted the immune function in laying hens as reported by Azzam et al. (2011a), Azzam et al. (2011b), and Dong et al. (2016) [3, 36, 37]. According to Dong et al. (2017), mucosal immunity was enhanced with increasing threonine [38]. These investigators suggested that when limiting amio acids in a low-crude protein diet of layers during peak production, threonine may be a limitig amino acid.

## Conclusion

In our work, production, egg quality, biochemical measurements, and ileal digestibility were most significantly affected by varying level of valine from 0.81 to 0.87%, when isoleucine was constant at 0.72%. Information from the present work is valuable for both researchers and producers. Results of our previous work, induducated that addition of isoleucine produced a low FCR at 0.72%. When investigating levels of isoluecne and valine in the present work with older lyers, FCR was reduced with 0.72% isolecine and 0.87% valine in the diet. More research will

narrow the range for appropriate quantities of valine when isoleucine is held at 0.72% in diets for hens in early stages of egg production. Work on comparison of all BCAA's should be performed on the same group of layers over their lifetime. In this way, use of BCAAs could be optimized for various genetic types throughout each period of production. Presently, producers can choose the amount of isolecucine (72%) and valine to maximize their most important commecial parameter(s).

## Materials and methods

### Diets, housing, and care

Layers (490, 33-weeks-old LSL-LITE starting weight of 1.581 ± 0.065 Kg) were randomly assigned to 7 dietary treatments × 7 replicates × 10 layers per replicate for 8 weeks. Mash diets with 0.72% isoleucine + 0.72, 0.75, 0.78, 0.81, 0.84, 0.87 and 0.90% total valine were formulated and prepared. The diet containing 0.72% of both isoleucine and valine was considered the Control. Several low cost ingredients were used to replace more expensive soy (protein source) in the diet (Table 5) [39–42]. Diet and fresh water were provided *ad libitum*.

The total quantity of protein from all sources and amino acids are shown in Table 6 (Amino LAB[R]) [43].

### Bird care, egg quality, and measurements

The Ethical Review Committee, University of Veterinary and Animal Sciences (Lahore, Pakistan) approved the protocol housing, feeding, and care of layers. Housing conditions included a caged system (60 x 63.5 cm, five birds each) on a commercial layer farm with uniform egg production and body weight, constant 16L:8D photoperiod (10–15 Lux), a maximum (26.68 ± 0.21 SE ˚C), and a minimum (21.75 ± 0.24 SE ˚C) temperature during October and November. All procedures and measurements were those noted for Ullah et al., 2021 [23].

### Statistical analysis

Data were subjected to a one way ANOVA under a completely randomized design (Steel et al., 1997) [44]. Weekly data for feed intake, FCR, body weight change, and egg quality parameters were analyzed by repeated measures SAS (version 9.1; SAS Inst. Inc. Cary, NC) using PROC GLM [45]. The model was $Y_{ijk} = \mu + \tau_i + P_j + (\tau \times P)_{ij} + \varepsilon_{ijk}$. where: $Y_{ijk} = Y$ is observation of $i^{th}$ treatment on $j^{th}$ period, $\mu$ = the overall mean, $\tau_i$ = the effect of $i^{th}$ treatment (i = 1, 2,3. ...7), $P_j$ = the effect of $j^{th}$ period (P = 1, 2, 3. . .8), $(\tau \times P)_{ik}$ = the effect of interaction between $i^{th}$ treatment and $j^{th}$ period, $\varepsilon_{ijk}$ = random error associated with $k^{th}$ observation on $j^{th}$ period on $i^{th}$ treatment NID ~ mean 0 and variance $\sigma^2$. Data were also evaluated using orthogonal

**Table 5. Protein content in 100g of various components in layer diets.**

| Component | Grams/100grams |
|---|---|
| Conventional canola meal[1] | 40.73 to 43.01 |
| De hulled sunflower seed meal[2] | 42.00 |
| Soybean meal[3] | 52.00 |
| Poultry byproduct meal[4] | 61.20 |

[1]Chen et al., 2015 [39].

[2]González-Pérez, 2015 [40].

[3]Banaszkiewicz, 2007 [41].

[4]Pesti, 1986 [42].

**Table 6. Feed and nutrient composition of the control diet.**

| Dietary Ingredients | % | Nutrients Composition | Calculated@ | Analyzed* |
|---|---|---|---|---|
| | | | —(%)——— | |
| Corn | 54 | Dry Matter | 90.36 | 90.25 |
| Rice Tips | 6.1 | Metabolizable Energy | 2730 | ————— |
| Canola Meal | 5.0 | Crude Protein | 16.70 | 17.21 |
| Sunflower seed meal | 5.0 | Ether Extract | 3.06 | ————— |
| Corn Gluten 60 | 2.0 | Crude Ash | 2.27 | ————— |
| Soybean Meal | 12 | Crude Fiber | 4.33 | ————— |
| Guar Meal | 2.0 | Calcium | 3.56 | ————— |
| Poultry By-Product Meal | 2.0 | Available phosphorus | 0.42 | ————— |
| Oil | 0.7 | Total phosphorus | 0.67 | ————— |
| $CaCO_3$ | 8.0 | Sodium | 0.17 | ————— |
| DCP | 1.8 | Potassium | 0.62 | ————— |
| Lysine-$SO_4$ | 0.4 | Chloride | 0.16 | ————— |
| DL-methionine | 0.15 | Dig[4] Lysine | 0.83 | 0.909 |
| L-Threonine | 0.1 | Dig Methionine | 0.42 | 0.303 |
| L-Tryptophan | 0.05 | Dig Threonine | 0.62 | 0.609 |
| L-Valine | 0.05 | Dig Tryptophan | 0.18 | 0.186 |
| L-Isoleucine[#] | 0.1 | Dig Cystine | 0.26 | 0.294 |
| $NaHCO_3$ | 0.4 | Dig M+C | 0.67 | 0.597 |
| Vitamin/Mineral Premix[##] | 0.3 | Dig Arginine | 0.95 | 1.11 |
| Total | 100 | Dig Valine | 0.72 | 0.864 |
| | | Dig Isoleucine | 0.73 | 0.764 |
| | | Dig Leucine | 1.29 | 1.464 |
| | | Dig Histidine | 0.37 | 0.427 |
| | | Dig Phenylalanine | 0.69 | 0.825 |
| | | Linoleic acid | 1.66 | ————— |
| | | Na+K-Cl (mEq / Kg) | 214.35 | ————— |

@Digestible amino acids values were calculated during feed formulation.

*Values are total digestible amino acids (Amino Lab® Evonik SEA Pte. Ltd. Singapore, lab code: SG16-0000618-001) for all treatments).

#Purity of L-isoleucine was > 98.0.

##Vitamin/mineral premix/kg of feed: Vitamin A, 10000 IU: Vitamin D3, 2500 IU: Vitamin E, 15–30 mg; Vitamin K3, 3 mg; Vitamin B1, 1 mg; Vitamin B2, 4 mg; Vitamin B6, 3mg; Vitamin B12, 15 mcg; Pantothenic Acid, 8 mg; Nicotinic Acid, 30 mg; Folic Acid, 1 mg; Biotin, 50 mcg; Choline, 300 mg; Manganese, 100 mg; Zinc, 60 mg; Iron, 25 mg; Copper, 5 mg; Cobalt, 0.1 mg; Iodine, 0.1 mg; Selenium, 0.2 mg.

polynomials for linear, quadratic, and cubic responses. Statistical significance at $P \leq 0.05$ and trends ($P \leq 0.10$) were determined using Duncan's Multiple Range Test (Duncan, 1955) [46].

## Acknowledgments

Authors thank the commercial layer farm for cooperation while conducting the research.

## Author Contributions

**Conceptualization:** Usman Liaqat, Yasir Ditta, Saima Naveed, Annie King, Talat Pasha, Khalid Abdul Majeed.

**Data curation:** Usman Liaqat, Yasir Ditta, Saima Naveed, Sana Ullah.

**Formal analysis:** Annie King.

**Investigation:** Khalid Abdul Majeed.

**Methodology:** Saima Naveed, Talat Pasha, Sana Ullah.

**Project administration:** Talat Pasha.

**Supervision:** Sana Ullah, Khalid Abdul Majeed.

**Visualization:** Annie King.

**Writing – review & editing:** Annie King.

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
