## [Decision Letter · Decision Letter 0]

12 Jan 2021

PONE-D-20-36625

Effect of L-valine in layer diets containing 0.72% isoleucine

PLOS ONE

Dear Dr. King,

Thank you for submitting your manuscript to PLOS ONE. After careful consideration, we feel that it has merit but does not fully meet PLOS ONE’s publication criteria as it currently stands. Therefore, we invite you to submit a revised version of the manuscript that addresses the points raised during the review process.

We look forward to receiving your revised manuscript.

Kind regards,

Mahmoud A.O. Dawood, PhD

Academic Editor

PLOS ONE

Journal Requirements:

2.We note that you have indicated that data from this study are available upon request. PLOS only allows data to be available upon request if there are legal or ethical restrictions on sharing data publicly. For more information on unacceptable data access restrictions, please see http://journals.plos.org/plosone/s/data-availability#loc-unacceptable-data-access-restrictions.

4. Please amend your authorship list in your manuscript file to include author Khalid Majeed.

"This research was supported by EVONIK Animal Nutrition, Singapore."

Additionally, because some of your funding information pertains to commercial funding, we ask you to provide an updated Competing Interests statement, declaring all sources of commercial funding.

In your Competing Interests statement, please confirm that your commercial funding does not alter your adherence to PLOS ONE Editorial policies and criteria by including the following statement: "This does not alter our adherence to PLOS ONE policies on sharing data and materials.” as detailed online in our guide for authors  http://journals.plos.org/plosone/s/competing-interests.  If this statement is not true and your adherence to PLOS policies on sharing data and materials is altered, please explain how.

Please include the updated Competing Interests Statement and Funding Statement in your cover letter. We will change the online submission form on your behalf

Reviewers' comments:

Reviewer's Responses to Questions

**Comments to the Author**

1. Is the manuscript technically sound, and do the data support the conclusions?

Reviewer #1: Yes

Reviewer #2: Partly

2. Has the statistical analysis been performed appropriately and rigorously? 

Reviewer #1: Yes

Reviewer #2: No

3. Have the authors made all data underlying the findings in their manuscript fully available?

Reviewer #1: Yes

Reviewer #2: Yes

4. Is the manuscript presented in an intelligible fashion and written in standard English?

Reviewer #1: Yes

Reviewer #2: No

5. Review Comments to the Author

Reviewer #1: Dear Authors

Regarding the manuscript title Effect of L-valine in layer diets containing 0.72% isoleucine

The scientific background of the topic was well mentioned in the introduction part. The experiment design, as well as the replicates and methods used, were very good. The results obtained were presented in tables well discussed with other author’s results. However, some observation in the present paper should be corrected and add to improve the quality of the paper.

Introduction

1- Need more information about the effects of isoleucine in layer hens.

2- Need recently references for using Valine in layer diets

Discussion

3- 3- In this part you focused on the studies which agree or disagree with your results without a)ny explanation why these findings were agree with disagree with you.

4- You must write the mode of action of L-valine and isoleucine on layer performance and egg quality

5- I advise to use some recently reference for using amino acids in poultry and the additives affect the egg quality and immunity

I recommend reading and using the following references:

Saleh A. A. (2016) Effect of Low-Protein in Iso-Energetic Diets on Performance, Carcass Characteristics, Digestibilities and Plasma Lipids of Broiler Chickens. Egyptian Poultry Science Journal. Vol (36) (I): (251-262).

Saleh A. A, Mohammed S. Eltantawy , Esraa M. Gawish , Hassan H. Younis , Khairy A. Amber , Abd El-Moneim E. Abd El-Moneim & Tarek A. Ebeid (2020) Impact of Dietary Organic Mineral Supplementation on Reproductive Performance, Egg Quality Characteristics, Lipid Oxidation, Ovarian Follicular Development, and Immune Response in Laying Hens Under High Ambient Temperature. Biological Trace Element Research. 195:506–514.

Saleh, AA., Ashia Zaki, Ahmed El- Awady, Khairy Amber, Neamat Badwi, Yahya Eid, Tarek A. Ebeid (2020) Effect of substituting wheat bran with cumin seed meal on laying performance, egg quality characteristics and fatty acid profile in laying hens. Veterinarski Arhiv 90 (1), 47-56,

Saleh A. A., Abeer A. Kirrella, Mahmoud A. O. Dawood, Tarek A. Ebeid (2019) Effect of dietary inclusion of cumin seed oil on the performance, egg quality, immune response and ovarian development in laying hens under high ambient temperature. Animal Physiology and Animal Nutrition. 103(6):1810-1817.

6- Material and Methods

Why didn’t you prepare your diets according the layer strain catalog (LSL-LITE)?

P13 L1 Mash and fresh water were provided ad libitum, what is mean of mash are you mean feed form?

Table 5. Feed and nutrient composition of diets.

You must provide the crude protein levels of (Canola Meal, Sunflower seed meal, Soybean Meal and Poultry By-Product Meal

Reviewer #2: This is the review for the manuscript PONE-D-20-35025, entitled “L-valine in diets with 0.72% isoleucine: effect on production, egg quality, serum biochemistry, and ileal digestibility of protein” for PLOS ONE Journal.

The manuscript aims at investigating the effect of dietary supplementation of L-valine on laying hen performance in terms of egg production and quality, serum biochemistry, and protein ileal digestibility.

I have the following more detailed comments that can help authors in the revision of the manuscript:

Page 4 (Table 1): Kindly check body weight values for laying hens. It seems not correct. These values more appropriate for egg weight not hen body weight.

Page 8 (Line 5): delete this sentence

Page 10 (Line 16): The effect of valine: lysine on egg production is not discussed. The authors only compared with the findings in the previous studies and nothing mentioned why this alteration has occurred.

Page 12 (Material and methods)

Page 12 (Line 19): 490 refers to what? Kindly write (N=490). Kindly provide the average initial hen weight.

Nothing mentioned about blood or serum collection or analysis, and ileal digestibility, egg quality assessment.

Kindly check the superscripts according to the pooled SE and presented values of mean through the whole manuscript

6. PLOS authors have the option to publish the peer review history of their article (what does this mean?). If published, this will include your full peer review and any attached files.

Reviewer #1: No

Reviewer #2: No

---

## [Author Response · Author response to Decision Letter 0]

10 Sep 2021

Reviewers' comments:

Reviewer's Responses to Questions

Comments to the Author

1. Is the manuscript technically sound, and do the data support the conclusions?

Reviewer #1: Yes

Reviewer #2: Partly

More information was provided throughout the manuscript to improve it.

2. Has the statistical analysis been performed appropriately and rigorously? 

Reviewer #1: Yes

Reviewer #2: No

Reviewer 2 did not provide information on where the analysis was weak. 

 We conducted regression analysis on each measurement to determine all responses as discussed in the manuscript.

3. Have the authors made all data underlying the findings in their manuscript fully available?

Reviewer #1: Yes

Reviewer #2: Yes

Data will be deposited at the UC Davis Library, University of California, Davis, CA 95616. 

4. Is the manuscript presented in an intelligible fashion and written in standard English?

Reviewer #1: Yes

Reviewer #2: No

No specific errors were noted.

5. Review Comments to the Author

Reviewer #1: Dear Authors

Regarding the manuscript title Effect of L-valine in layer diets containing 0.72% isoleucine

The scientific background of the topic was well mentioned in the introduction part. The experiment design, as well as the replicates and methods used, were very good. The results obtained were presented in tables well discussed with other author’s results. However, some observation in the present paper should be corrected and add to improve the quality of the paper.

More information has been added to all parts of the manuscript. In many places, it was totally rewritten. 

Introduction

1- Need more information about the effects of isoleucine in layer hens.

2- Need recently references for using Valine in layer diets

More information was added to the Introduction.

Discussion

3- 3- In this part you focused on the studies which agree or disagree with your results without a)ny explanation why these findings were agree with disagree with you.

More discussion was added as advised.

4- You must write the mode of action of L-valine and isoleucine on layer performance and egg quality.

More discussion was added in the Introduction and Results and Discussion.

5- I advise to use some recently reference for using amino acids in poultry and the additives affect the egg quality and immunity

I recommend reading and using the following references:

Saleh A. A. (2016) Effect of Low-Protein in Iso-Energetic Diets on Performance, Carcass Characteristics, Digestibilities and Plasma Lipids of Broiler Chickens. Egyptian Poultry Science Journal. Vol (36) (I): (251-262).

Saleh A. A, Mohammed S. Eltantawy , Esraa M. Gawish , Hassan H. Younis , Khairy A. Amber , Abd El-Moneim E. Abd El-Moneim & Tarek A. Ebeid (2020) Impact of Dietary Organic Mineral Supplementation on Reproductive Performance, Egg Quality Characteristics, Lipid Oxidation, Ovarian Follicular Development, and Immune Response in Laying Hens Under High Ambient Temperature. Biological Trace Element Research. 195:506–514.

Saleh, AA., Ashia Zaki, Ahmed El- Awady, Khairy Amber, Neamat Badwi, Yahya Eid, Tarek A. Ebeid (2020) Effect of substituting wheat bran with cumin seed meal on laying performance, egg quality characteristics and fatty acid profile in laying hens. Veterinarski Arhiv 90 (1), 47-56,

Saleh A. A., Abeer A. Kirrella, Mahmoud A. O. Dawood, Tarek A. Ebeid (2019) Effect of dietary inclusion of cumin seed oil on the performance, egg quality, immune response and ovarian development in laying hens under high ambient temperature. Animal Physiology and Animal Nutrition. 103(6):1810-1817.

Many of these papers had to be requested from the authors or there was a monetary fee attached to them. The authors thank the Reviewer for the comment and found other useful articles. 

6- Material and Methods

Why didn’t you prepare your diets according the layer strain catalog (LSL-LITE)?

In Material and Methods, we explained that other available and less expensive sources of protein were used to replace soy. We provided an evaluation of protein (and references) for the sources. It is not reasonable to assume that all researchers throughout the world can use the exact same diets used by the producers of the layers. 

P13 L1 Mash and fresh water were provided ad libitum, what is mean of mash are you mean feed form? 

We have explained that "mash" was the form of the diet. This has not been an issue with other manuscripts.

Table 5. Feed and nutrient composition of diets.

You must provide the crude protein levels of (Canola Meal, Sunflower seed meal, Soybean Meal and Poultry By-Product Meal

These were provided in Table 5 (Materials and Methods).

Reviewer #2: This is the review for the manuscript PONE-D-20-35025, entitled “L-valine in diets with 0.72% isoleucine: effect on production, egg quality, serum biochemistry, and ileal digestibility of protein” for PLOS ONE Journal.

The manuscript aims at investigating the effect of dietary supplementation of L-valine on laying hen performance in terms of egg production and quality, serum biochemistry, and protein ileal digestibility.

I have the following more detailed comments that can help authors in the revision of the manuscript:

We thank Reviewer # 2 for the comments.

Page 4 (Table 1): Kindly check body weight values for laying hens. It seems not correct. These values more appropriate for egg weight not hen body weight. 

The information was removed.

Page 8 (Line 5): delete this sentence

It was deleted.

Page 10 (Line 16): The effect of valine: lysine on egg production is not discussed. The authors only compared with the findings in the previous studies and nothing mentioned why this alteration has occurred. 

More information was added.

Page 12 (Material and methods)

Page 12 (Line 19): 490 refers to what? Kindly write (N=490). Kindly provide the average initial hen weight.

Please see Material and Methods - Diets, Housing and Care.

Nothing mentioned about blood or serum collection or analysis, and ileal digestibility, egg quality assessment.

This information is included in the first paper on isoleucine. We are also submitting revisions for this work, “Varying digestible isoleucine level to determine effects on performance, egg quality, serum biochemistry, and ileal protein digestibility in diets of young laying hens” and hope to have it be approved for publication and published before this present manuscript. Revisions for the manuscript on work with isoleucine should have been completed first; however, communication between colleagues in Pakistan and the US was delayed for some time.

_________

Kindly check the superscripts according to the pooled SE and presented values of mean through the whole manuscript.

These have been added throughout the Results and Discussion.

6. PLOS authors have the option to publish the peer review history of their article (what does this mean?). If published, this will include your full peer review and any attached files.

Yes.

---

## [Editor Report · Decision Letter 1]

23 Sep 2021

Effects of L-valine in layer diets containing 0.72% isoleucine

PONE-D-20-36625R1

Dear Dr. King,

We’re pleased to inform you that your manuscript has been judged scientifically suitable for publication and will be formally accepted for publication once it meets all outstanding technical requirements.

Kind regards,

Mahmoud A.O. Dawood, PhD

Academic Editor

PLOS ONE
---

## [Editor Report · Acceptance letter]

9 Dec 2021

PONE-D-20-36625R1 

Effects of L-valine in layer diets containing 0.72% isoleucine 

Dear Dr. King:

I'm pleased to inform you that your manuscript has been deemed suitable for publication in PLOS ONE. Congratulations! Your manuscript is now with our production department. 

Kind regards, 

on behalf of

Dr. Mahmoud A.O. Dawood 

Academic Editor

PLOS ONE